# RAS Mediates BET Inhibitor-Endued Repression of Lymphoma Migration and Prognosticates a Novel Proteomics-Based Subgroup of DLBCL through Its Negative Regulator IQGAP3

**DOI:** 10.3390/cancers13195024

**Published:** 2021-10-07

**Authors:** Chih-Cheng Chen, Chia-Chen Hsu, Sung-Lin Chen, Po-Han Lin, Ju-Pei Chen, Yi-Ru Pan, Cih-En Huang, Ying-Ju Chen, Yi-Yang Chen, Yu-Ying Wu, Muh-Hwa Yang

**Affiliations:** 1Division of Hematology and Oncology, Department of Medicine, Chang Gung Memorial Hospital, Chiayi 61363, Taiwan; ccchen@cgmh.org.tw (C.-C.C.); loofah1008@cgmh.org.tw (C.-C.H.); 8802058@cgmh.org.tw (C.-E.H.); agrb7289@cgu.edu.tw (Y.-J.C.); greatspirit0406@gmail.com (Y.-Y.C.); yuyun5801@cgmh.org.tw (Y.-Y.W.); 2School of Medicine, Chang Gung University, Taoyuan 33302, Taiwan; 3Institute of Biotechnology in Medicine, National Yang Ming Chiao Tung University, Taipei 11221, Taiwan; jason840714@gmail.com; 4Institute of Clinical Medicine, National Yang Ming Chiao Tung University, Taipei 11221, Taiwan; phlin@nycu.edu.tw; 5Cancer Progression Research Center, National Yang Ming Chiao Tung University, Taipei 11221, Taiwan; pei03231987@nycu.edu.tw; 6Division of General Surgery, Department of Surgery, Chang Gung Memorial Hospital, Taoyuan 33302, Taiwan; panyiru0331@cgmh.org.tw; 7Division of Medical Oncology, Department of Oncology, Taipei Veterans General Hospital, Taipei 11217, Taiwan

**Keywords:** BET inhibition, diffuse large B-cell lymphoma, RAS GTPase, Rho GTPase, IQGAP3, PI3K, migration

## Abstract

**Simple Summary:**

The inhibitors of BET proteins represent a promising class of therapeutic agents that target the oncogenic activity of MYC and repress DLBCL cell migration, but the mechanism of such repression remains elusive. Herein, we found that BET inhibitor JQ1 abrogated the amoeboid movement of DLBCL cells through a small GTPase-driven mechanism, including both restrained RAS signaling and MYC-mediated suppression of GTP-RhoA activity. BET inhibition drastically increased the expression of a GTPase regulatory protein, the IQ motif containing GTPase activating protein 3 (IQGAP3), in DLBCL. Proteomics-based re-stratification identified a specific subgroup of DLBCL patients whose tumors harbored an enhanced PI3K activity and had an inferior survival, whereas a lower *IQGAP3* expression level further portended a very dismal outcome for those patients. The inhibitors of both BET and RAS (through attenuated PI3K signaling) activities effectively ameliorated the outspread of in vivo DLBCL tumors, indicating the potential of their synergism in the treatment of specific DLBCL subtypes.

**Abstract:**

Phenotypic heterogeneity and molecular diversity make diffuse large B-cell lymphoma (DLBCL) a challenging disease. We recently illustrated that amoeboid movement plays an indispensable role in DLBCL dissemination and inadvertently identified that the inhibitor of bromodomain and extra-terminal (BET) proteins JQ1 could repress DLBCL migration. To explore further, we dissected the impacts of BET inhibition in DLBCL. We found that JQ1 abrogated amoeboid movement of DLBCL cells through both restraining RAS signaling and suppressing MYC-mediated RhoA activity. We also demonstrated that BET inhibition resulted in the upregulation of a GTPase regulatory protein, the IQ motif containing GTPase activating protein 3 (IQGAP3). IQGAP3 similarly exhibited an inhibitory effect on RAS activity in DLBCL cells. Through barcoded mRNA/protein profiling in clinical samples, we identified a specific subgroup of DLBCL tumors with enhanced phosphatidylinositol-3-kinase (PI3K) activity, which led to an inferior survival in these patients. Strikingly, a lower *IQGAP3* expression level further portended those with PI3K-activated DLBCL a very dismal outcome. The inhibition of BET and PI3K signaling activity led to effective suppression of DLBCL dissemination in vivo. Our study provides an important insight into the ongoing efforts of targeting BET proteins as a therapeutic approach for DLBCL.

## 1. Introduction

Diffuse large B-cell lymphoma (DLBCL) is the most common lymphoid malignancy in adulthood worldwide. Although durable remission can be achieved with combination immunochemotherapy, about one-third of the patients with DLBCL develop recurrent or progressive diseases that are highly fatal [1]. The striking heterogeneity of the disease has long been considered the major culprit of discrepant treatment outcome in those patients, and more effective strategies targeting the pathognomonic mechanism of certain difficult to treat lymphoma subtypes are urgently needed. 

Studies have shown that DLBCL cells frequently harbor mutations that perturb epigenetic regulation and lead to neoplastic transformation of B cells [2,3]. Consequently, epigenetics-based therapeutics represents a rational approach in efforts to exploit novel agents for DLBCL. Among all, the members of the bromodomain and extra-terminal (BET) family of proteins have become appealing targets. BET proteins play a crucial role as epigenetic readers [4] through binding to acetylated lysine residues on histone tails to modulate the expression of key genes defining the biology of normal cells and of oncogenes engaging in tumorigenesis [4,5,6]. Recently, several small-molecule inhibitors have been developed to competitively interfere with the histone-binding capacity and epigenetic activity of BET proteins [7,8]. Through targeting of MYC dependence in cancer cells [7,9], they have shown promising efficacy in a variety of solid tumors and hematological malignancies [5], including DLBCL [8,10]. 

The mechanism of action of BET inhibitors in DLBCL has been explored thoroughly. Except for the known effects on MYC modulation, BET inhibitors might as well attenuate the activities of signaling pathways such as nuclear factor NF-κB, and Janus kinase/signal transducers and activators of transcription (JAK/STAT) that are critically involved in lymphomagenesis [10,11,12]. Furthermore, BET inhibitors also alter the expression of pathologically relevant microRNAs in DLBCL [13]. These findings provide a mechanistic basis for the employment of BET inhibitors in DLBCL and lead to success in an early phase clinical trial [14]. 

Amoeboid movement, characterized by migration with a low adhesion force and high actomyosin-mediated contractility, is commonly observed in the progression of different types of cancers, including lymphoma [15]. In our recent work, we demonstrated that STAT3-coordinated amoeboid movement led to the dissemination of DLBCL cells, and strategies aiming at suppressing DLBCL cell migration significantly reduced the tumor burden in an in vivo xenograft model [16]. Specifically, in an effort to identify effective agents that suppressed the motility of DLBCL cells, we inadvertently recognized that the BET inhibitor JQ1 also reduced amoeboid movement [16]. This unexpected finding raises the possibility of the presence of some unidentified mechanism of action related to BET inhibition in DLBCL. 

Here, we explore through the molecular pathogenesis of BET inhibition in DLBCL. Mechanistic studies uncover dual suppressive effects on GTP-RhoA and GTP-RAS by BET inhibitors, whereas the expression of a GTPase-regulating protein is found to be of clinical significance in a cohort of DLBCL patients. These findings provide a scientific rationale in incorporating BET inhibitors into the treatment paradigm in DLBCL.

## 2. Materials and Methods

### 2.1. Trajectory Tracking of DLBCL Cell Migration

For tracking DLBCL migration, we deployed 200 µL of 1.5 mg/mL collagen solution (Advanced BioMatrix, Inc. Carlsbad, CA, USA) in the lower layer and DLBDL cells were suspended in 200 µL of 1.5 mg/mL collagen solution in the upper layer. After solidification of the collagen solution, the inhibitor/vehicle control-containing medium were added and incubated for 6 or 24 hours. Information on tracking and image capture is listed in the Appendix A.

### 2.2. Small GTPase Activity Assay and Immunoblotting

For assaying Ras, RhoA, Cdc42, Rac1 activity, the DLBCL cells were stimulated with additional 10% FBS (total 20%) for 1 day to increase the level of active GTPases. Following treatment with indicated inhibitors, the cells were lysed and subjected to GST-immobilization targeted pull down and immunoblotting analysis. Other details are depicted in the Appendix A. All original western blot figures can be found in Appendix A.

### 2.3. Mouse Experiments

All processes were guided in accordance with the institutional animal welfare guidelines of the Taipei Veteran General Hospital (IACUC No.2016-037) and Chang-Gung Memorial Hospital, Chiayi (IACUC No.2019122014). CB17.Cg-*Prkdc^scid^Lyst^bg-J^*/CrlBltw (SCID/beige) mice were purchased from BioLASCO Taiwan Co., Ltd., Taipei, Taiwan. Mouse experiments in Appendix A are described in the Appendix A. 

### 2.4. Patients and Sample Collection

We retrospectively enrolled 53 patients with DLBCLs who were treated with immunochemotherapy at our institute. In addition, six healthy adults were included as controls. The study was approved by the Institutional Review Board of Chang-Gung Memorial Hospital. Please refer to the Appendix A for the demographics of the patients and details on sample collection.

### 2.5. Barcoded mRNA and Protein Profiling with the nCounter Technology

Digital gene expression profiling (GEP) was performed to assess the expression of 240 genes using a barcoded profiling method (nCounter technology, NanoString Technologies, Inc., Seattle, WA, USA). The barcoded protein profiling was performed with the nCounter^®^ Vantage 3D™ Protein Panel, which detected 35 target proteins in one formalin-fixed paraffin-embedded (FFPE) slide. Additional details are provided in the Appendix A. 

### 2.6. External Validation

For independent external validation, we obtained a gene expression data set from published DLBCL mRNA series by Lenz et al. (GSE11318) for comparison [17]. Information on this dataset and relevant statistics are provided in the Appendix A. 

### 2.7. Others

Information on cell culture, plasmids, virus infection, quantitative RT-PCR (RT-qPCR), flow cytometry, high-throughput RNA sequencing, and statistics are all listed in the Appendix A.

## 3. Results

### 3.1. BET Inhibitors Suppresses Amoeboid Movement and Dissemination of DLBCL Cells

We previously demonstrated that the BET inhibitor JQ1 could suppress the amoeboid movement of SU-DHL-5 cells [16]. To validate this finding, we obtained two BET inhibitors (JQ1 and AZD5153) and three DLBCL cell lines (SU-DHL-5, HT, and Toledo) to repeat the experiment. When added to cells deployed in the collagen solution (Figure 1a), both JQ1 and AZD5153 significantly reduced the migration of DLBCL cells (Figure 1b,c). To avoid the potential interference of excessive apoptosis on gauging the cellular movement, we also assessed these two agents in relevant assays. Although treatment with the two inhibitors at the doses of migration assay resulted in partial inhibition of retinoblastoma (Rb) protein phosphorylation (Appendix A, left panel), flow cytometry nevertheless confirmed that the percentage of sub-G1 cell population remained stable (Appendix A). The cellular viability was not significantly affected by both, either (Appendix A). The minimal impacts of both BET inhibitors at the experimental doses on cellular apoptosis could also be demonstrated on Annexin V staining (Appendix A) and measurement of cleaved caspase 3 (Appendix A). Based on the results, we confirmed that, at sublethal doses, BET inhibitors suppressed DLBCL migration without inducing apoptosis during the assay.

To further scrutinize the effects of BET inhibition on tumor dissemination, we next assessed the growth of DLBCL cells in an orthotopic xenograft model. We previously cultivated derivative sublines HT-1 and HT-L1 from HT cells through inoculation of the parental cells into subcutaneous regions of SCID/beige mice [16]. Both sublines exhibited enhanced in vivo tumorigenicity [16]. In the current work, experimental mice received intrasplenic injection of HT-L1 cells carrying an iRFP713 fluorescent reporter (HT-L1-iRFP), and JQ1 was administered intraperitoneally starting at the end of week three (Figure 1d– e). JQ1 treatment at the dose of 25 mg/kg twice weekly significantly impaired the in vivo dissemination of DLBCL cells (Figure 1f), a result confirmed on both the enumeration of outgrowing fluorescent tumor spots (Figure 1g) and the quantification of overall near infrared signals (Figure 1h). These data suggest that BET inhibition suppresses the amoeboid movement and dissemination of DLBCL cells.

### 3.2. JQ1-Mediated Restraint of DLBCL Migration Is Partially Attributable to the Inactivated MYC-RhoA Loop 

MYC-orchestrated RhoA transcription has been shown to be pivotal in cell invasion and cancer metastasis [18], and MYC-regulated genes are one of the major targets held responsible for the antiproliferative effects of BET inhibitors in DLBCL [11,12,19]. Furthermore, we have also demonstrated that RhoA indispensably contributes to DLBCL migration [16]. Since MYC is the identified primary target of BET inhibitors, we first looked into the activity of MYC-RhoA axis in JQ1-treated DLBCL cells to explore the major mechanism involved in BET inhibition-mediated suppression of amoeboid movement in DLBCL. The downregulation of MYC mRNA and protein could be consistently observed across all five DLBCL lines (SU-DHL-5, HT, Toledo, LY3, and U2932) treated with JQ1 (Figure 2a). We next investigated whether BET inhibition affected the small Rho GTPases activity. Using a glutathione S-transferase (GST) pull-down assay for detecting small GTPase activity, we found that JQ1 ameliorated GTP-RhoA activity in several DLBCL cell lines, whereas the total RhoA protein levels remained stable in the treated cells (Figure 2b). On the other hand, the activities of GTP-Rac1 and GTP-Cdc42, the other two key members of Rho GTPase subfamilies [20], were not significantly altered by JQ1. 

Subsequently, we downregulated the expression of MYC with short hairpin RNAs (shRNA) and found that the GTP-RhoA activity was repressed in Toledo cells (Figure 2c). This indicates that JQ1-induced suppression of RhoA activity is probably mediated through MYC inhibition. With the movement of DLBCL cells constrained by BET inhibition, we wondered if MYC abrogation would lead to similar effects. Consistently, decreased MYC expression by shRNAs mitigated the motility of DLBCL cells (Figure 2d). Importantly, when combined with JQ1 treatment, MYC depletion further diminished the migration of DLBCL cells (Figure 2e). These data imply that BET inhibition leads to MYC downregulation and resultant repression of GTP-RhoA activity. However, the synergy between MYC depletion and JQ1 inhibition in abolishing cellular movement also alludes to potentially unidentified mechanisms other than the MYC-RhoA axis in the JQ1-endued suppression of DLBCL migration.

### 3.3. BET Inhibition Also Leads to Attenuated RAS Activity in DLBCL Cells

To comprehensively delineate the molecular mechanism of BET inhibitor-associated suppression of migration, we examined the impact of BET inhibition on the transcriptomic profiling of DLBCL cells. We performed RNA sequencing in SU-DHL-5 and Toledo cells treated with either DMSO or JQ1. Using 0.5 as the cutoff ratio comparing gene expression levels between differentially treated cells, we found that 3,768 and 4,139 genes were significantly downregulated in JQ1-treated SU-DHL-5 and Toledo cells, respectively (Figure 3a; Appendix A). Among them, there were 1,809 overlapping genes (Appendix A). By employing a Gene Ontology pathway analysis, we identified several key signaling signatures that were differentially inactivated in JQ1-treated cells (Figure 3b; Appendix A). Remarkably, small GTPase-mediated signal transduction signatures, especially RAS signaling activity, were most significantly repressed after JQ1 treatment. With RAS-transformed epithelial cancer cells having been shown to exhibit amoeboid type morphological changes [21], we put our focus on this specific pathway. Using the small GTPase pull-down assay, we found that BET inhibition through either JQ1 (Figure 3c) or AZD5153 (Figure 3d) both led to reduced RAS activity. The total RAS protein levels, however, were not affected. To further delineate a RAS-mediated DLBCL phenotype, we next explored the effects of RAS repression in the migration assay. RAF-MEK (mitogen-activated protein kinase kinase) is the major downstream effector pathway of RAS signaling [22]. With direct RAS suppression infeasible, we selected MEK inhibitor trametinib for current investigation. Trametinib significantly constrained the motility of DLBCL cells (Figure 3e), indicating the crucial role of RAS activity in DLBCL migration. 

RAS proteins are small GTPase proteins in which the activities are switched between active GTP-bound and inactive GDP-bound conformations [23]. The functional effects of small GTPase are induced by guanine nucleotide exchange factors (GEFs) and inactivated by GTPase-activating proteins (GAPs) [23]. JQ1 competitively inhibits the binding of BET proteins and suppresses the expression of genes with super-enhancers [6,8]. We wondered if JQ1-induced repression of RAS activity was mediated through altered expression of RASGEFs or RASGAPs. Therefore, we examined their expression levels through RNA-sequencing and quantitative real-time PCR (qPCR) assays. In JQ1-treated DLBCL cells, key genes involved in RASGEF activities showed erratic expression patterns (Appendix A). To justify the findings, we explored through the clinical data from our cohort of DLBCL patients. We assessed the expression of genes and proteins in their FFPE samples using the nCounter profiling method (Figure 3f). As demonstrated in Figure 3g, some *RASGEF* genes were aberrantly expressed. However, when analyzed in the context of survival outcome, none of these genes foretold a unique prognostic pattern (Appendix A). Taken together, our data indicate that, although BET inhibition does attenuate RAS activity in DLBCL cells, *RASGEF* genes are probably inconsequential in mediating such an effect.

### 3.4. IQGAP3 Induces RAS Inactivation and Holds Prognostic Relevance in DLBCL

With *RASGEF* genes being irrelevant to BET inhibitor-mediated RAS suppression, we next scrutinized the expressional status of RASGAPs and other small GTPase-activating regulatory proteins in DLBCL. Among all, *IQ Motif Containing GTPase Activating Protein*
*3* (*IQGAP3*) was identified as the most prominently overexpressed gene in our clinical samples (Figure 4a). Consistently, *IQGAP3* was among several *RASGAP* genes that were peculiarly upregulated in JQ1-treated DLBCL cells (Appendix A). IQGAP3 belongs to the IQGAP family of proteins that act as scaffolds for various signaling pathways and that regulate diverse biological activities [24]. Although the exact function of IQGAP3 remains largely unexplored, its aberrant expression has been reported in several types of cancers [25,26]. Compared with normal controls, many of our DLBCL samples harbored overexpressed *IQGAP3*. Using the average expression level as the cutoff point, we then stratified our DLBCL patients into two subgroups. Strikingly, those with high *IQGAP3* expression had significantly better clinical outcome exhibited by longer progression-free (PFS) and overall survival (OS) (Figure 4b). With the survival curve reaching a smooth plateau after 10 years, it is suggested that more than 90% of the high *IQGAP3* expressors are probably cured with the standard immunochemotherapy. On the other hand, none of the other examined *RASGAP* genes portended prognostic significance on survival outcome in DLBCL patients (Appendix A). For further affirmation of our findings, the published DLBCL mRNA data set by Lenz et al. (GSE11318) was used for external validation [17]. Consistent with our finding, we identified that, in this cohort of 200 patients, those with an *IQGAP3* expression level higher than average had a significantly better survival outcome than their counterparts (Figure 4c, left panel). To stringently define the prognostic value of *IQGAP3*, we further excluded 78 patients whose expression levels were within the range of average ± 0.5 x standard deviation from this cohort. As shown on the right panel of Figure 4c, higher *IQGAP3* expression still portended an excellent prognosis in DLBCL patients. The results unequivocally validate the prognostic significance of *IQGAP3* expressional level in DLBCL. 

We next investigated the effects of JQ1 on the expressional status of *IQGAP3*. Among most of the DLBCL cells examined, treatment with JQ1 led to increased *IQGAP3* mRNA levels (Figure 4d), and the result was confirmed on protein-based analysis (Figure 4e). Importantly, shRNA-directed downregulation of *IQGAP3* resulted in enhanced RAS activities (Figure 4f). Overall, these findings indicate that BET inhibitors would upregulate *IQGAP3*, which in turn suppresses RAS activity. *IQGAP3* might play a preeminently novel role in prognosticating DLBCL patients. Nevertheless, whether inhibitors of BET proteins directly restrict RAS activity or indirectly mediate through IQGAP3 to repress RAS remains to be elucidated.

### 3.5. Proteomics-Based Re-Categorization Identified a Subgroup of Patients with PI3K-Activated DLBCL Whose Prognosis Could Be Further Defined by IQGAP3

The identification of recurrent mutations, chromosomal translocations, and somatic copy number variations through comprehensive genetic analysis has led to the re-stratification of distinct DLBCL subgroups with unique genotypic, epigenetic, and clinical characteristics [27,28,29,30]. To put the transcriptomic information and signal transduction activities into the perspectives of subtype classification, we scrutinized the GEP pattern in our DLBCL patients. We employed the barcoded nCounter mRNA and protein analysis in clinical FFPE samples. Traditionally, DLBCL can be categorized into activated B-cell-like (ABC) and germinal center B-cell-like (GCB) subtypes based on their cells of origin (COO) stratified by the GEP pattern [31,32], with the former representing the more aggressive form of the disease [32,33]. Using the same algorithm, we sub-classified our patients into ABC and GCB subtypes (Figure 5a). Patients with ABC DLBCL unequivocally exhibited an inferior survival outcome (Figure 5b). We tried to incorporate the prognostic value of *IQGAP3* into the COO classification to see if *IQGAP3* was particularly relevant in a specific genetic subtype. However, as shown in Figure 5c, *IQGAP3* lost its prognostic impact in respective ABC and GCB subtypes of DLBCL. 

One of the major advantages of nCounter analysis is the potential to quantify phosphorylated proteins in tumor samples, which provides important insights into the information on signal pathway activation. Using the nCounter HEME protein panel, we performed unsupervised proteomics clustering analysis with Pearson’s correlation. The clustering results showed that our cohort of DLBCL patients could be segregated into two distinct groups (A and B, Figure 5d). Although it seemed that the newly categorized group B contained more ABC tumors (Figure 5d), this novel stratification was actually irrelevant to COO on statistical analysis. To dig further, we found that, compared with their counterpart, group B DLBCL tumors exhibited more significantly increased activation and/or higher expression of phosphatidyl-inositol-3 kinase (PI3K) pathway-associated proteins, including PIK3CD, p-PDK1, p-AKT, p-IKK A/B, NF-κB, MYC, BCL2L1, and MCL1 (Figure 5e). Extraordinarily, this molecular pathway-annotated categorization held important prognostic impacts, as patients stratified in group B had a considerably shorter PFS (Figure 5f). As mentioned earlier, our cell line data demonstrated that IQGAP3 could suppress RAS activity (Figure 4f). Reports have shown that RAS and PI3K pathways might intersect to regulate each other and co-regulate downstream functions and that such a cross-talk actually engages in the development and evolution of cancers [34]. This instigated our attempt in dissecting a potential role of *IQGAP3* in DLBCL tumors with upregulated PI3K pathway activity. In patients whose lymphoma cells lacked activated PI3K signaling (group A), the expression level of *IQGAP3* did not harbinger any survival outcome implication (Figure 5g). On the contrary, higher *IQGAP3* level prognosticated a significantly superior clinical outcome in group B patients (Figure 5h). Specifically, increased *IQGAP3* expression conferred DLBCL patients with PI3K-activated tumors exceptional long-term survival, and these patients actually fared as well as those in group A (Figure 5f–h). The data also alluded to the fact that the prognosis in patients with PI3K-activated, *IQGAP3*-downregulated DLBCL was very dismal. These results suggest that PI3K signaling activation characterizes a specific subpopulation of patients with DLBLC whose prognosis could be further defined by *IQGAP3*, as its expressional level demarcates RAS activity in lymphoma cells. Specifically, high *IQGAP3* expression forebodes a remarkably excellent clinical outcome in patients with PI3K-activated DLBCL.

### 3.6. Inhibition of PI3K Signaling Similarly Represses In Vitro Migration and In Vivo Dissemination of DLBCL

With PI3K-activated DLBCL patients (group B in our cohort) manifesting inferior survival outcome, we wonder whether inhibition of the PI3K signaling activity could also alter the phenotypic characteristics of DLBCL. The PI3K inhibitor copanlisib was, therefore, employed for the subsequent experiments. As demonstration in Figure 6a, the movement of three different DLBCL cells (SU-DHL-5, HT, and Toledo) was drastically impeded upon treatment with copanlisib. While the addition of copanlisib effectively diminished downstream AKT phosphorylation (Appendix A), the selected dose of this agent for migration assay had minimal effects on the percentage of sub-G1 population (Appendix A), cellular viability (Appendix A), and apoptosis (Appendix A). Similar to the findings on BET and MEK inhibitors, the results suggest that copanlisib at the sublethal dose attenuates the in vitro migratory activity of DLBCL cells. 

To further elaborate on DLBCL dissemination, we also exploited in vivo studies to thoroughly delineate the impacts of PI3K inhibition on tumor spreading (Figure 6b–e). Experimental mice received intrasplenic injection of HT-L1-iRFP cells, and drugs were administered intraperitoneally starting at the end of week three (Figure 6b). Similar to previous experiment with JQ1, treatment with AZD5153 at the dose of 25 mg/kg twice weekly significantly impaired the in vivo dissemination of DLBCL cells compared with untreated control (Figure 6c, middle and left panels). Combined treatment with low dose copanlisib and reduced dose of AZD5153 effectively suppressed the outspread of DLBCL tumors as well (Figure 6c, right panel). These results were confirmed on both the enumeration of outgrowing fluorescent tumor spots (Figure 6d) and the quantification of overall near infrared signals (Figure 6e). Collectively, these data suggest that PI3K inhibitor copanlisib beneficially mitigates the migration and dissemination of DLBCL cells similar to the way BET inhibitors do.

Here, we summarize our findings by schematic illustration in Figure 7. BET inhibitors suppress DLBCL migration through the inhibition of two pathways: the MYC-RhoA and RAS-MEK pathways. Mechanistically, BET inhibitors repress MYC transcription to attenuate RhoA activity; the inhibitors also induce IQGAP3, which subsequently inactivates RAS. In DLBCL patients, IQGAP3 prognosticates a worse survival in PI3K-activated cases. A combination of the PI3K and BET inhibitors effectively restrains DLBCL dissemination.

## 4. Discussion

The MYC oncogene has long been an attractive target in cancer therapeutics. However, direct MYC targeting is challenging owing to its essential roles in developmental biology. BET inhibition is considered to hold the greatest promise as an alternative approach to indirectly abrogate MYC oncogenic activities [35]. Mechanistic studies have shown that it acts through anti-proliferation and apoptosis induction in DLBCL cells to achieve the desirable effects [10,11,36]. In this study, we take a step forward by illustrating novel mechanisms of BET inhibition in the suppression of DLBCL. We have previously demonstrated cytokine-mediated, signal transducer and activator of transcription 3 (STAT3)-coordinated amoeboid migration as the culprit of lymphoma dissemination, and combining a JAK inhibitor with a microtubule stabilizer could serve as a mechanism-derived therapeutic approach in DLBCL [16]. The present work highlights that the motility of DLBCL can as well be driven by an alternative pathway, namely the small GTPase-mediated movement. With BET inhibitors being able to sturdily repress both RAS and RhoA activities and with the fact that amoeboid movement facilitates the dissemination of DLBCL [16], it is plausible that combinative strategies tackling the mechanism adopted by lymphoma cells for their propagation may further improve the treatment outcome in these patients.

Our study also calls attention to the importance of subtype-driven, individualized therapeutic strategy in DLBCL. Recently, through comprehensive genetic analysis, DLBCL has been re-stratified into distinct subtypes with unique clinical features and molecular characteristics [27,28]. In the study by Chapuy et al., five groups of DLBCL tumors with coordinate genetic signatures were identified, with the cluster 5 subset being shown to harbor upregulated PI3K signaling activity, to exhibit enrichment in ABC tumors, and to portend a dismal prognosis [28]. Our group B patients, clustered in the novel pathway activity-categorized stratification, demonstrated exactly the same features. Nevertheless, we also found that the behavior of PI3K-activated DLBCL was not unanimous, as IQGAP3 further stratified this group of patients unequivocally. The poor prognosis of these PI3K-activated tumors could probably be averted by increased *IQGAP3* expression, since our in vitro model demonstrated that high IQGAP3 attenuated RAS activity. RAS is a small GTPase with oncogenic implication that has been closely related to a PI3K signaling pathway. As a result, diminished RAS activity might counteract the detrimental effects of activated PI3K signaling. The theory was vividly validated in our clinical cohort, as DLBCL patients with concurrent PI3K activation and *IQGAP3* upregulation actually had an exceptionally excellent survival outcome. This is of ultimate importance, since those patients would be otherwise deemed to have little chance of long-term survival when high PI3K activity is used as the sole prognostic factor. The clinical implication is even more significant, as such patients might be erroneously given more intensive treatment by their physicians to stave off a gloomy fate that never exists. 

Among the IQGAP family of proteins, the functional activities of IQGAP3 are the least explored. The closely related and highly homologous IQGAP1 protein has been shown to scaffold both the Ras-ERK and PI3K-Akt pathways [37]. Limited data implicate that IQGAP3 effects through Rac1 and Cdc42 [38], and salient functions of IQGAP3 include the regulation of cell proliferation and motility [24]. While our finding on IQGAP3 is novel, its prognostic relevance in DLBCL is rightfully unambiguous. Attenuated RAS signaling was uncovered through in vitro BET inhibition, which prompted us to explore through small GTPase activity regulators and identified the key role of IQGAP3 in clinical patients. IQGAP3, of which the expression could be upregulated by BET inhibition, constrained RAS signaling and specifically prognosticated a subgroup of DLBCL patients whose tumors harbored enhanced PI3K activity. It is not immediately clear whether IQGAP3 acts through its direct GTPase activity to affect RAS, but its strong prognostic significance is unequivocal. Theoretically, based on these findings, either BET or PI3K inhibition would provide substantial benefits in patients with activated PI3K signaling and depressed IQGAP3. As mentioned earlier, RAS signaling through PI3K pathway indispensably contributes to tumor initiation, motility, and invasiveness [39,40]. Therefore, on top of PI3K inhibition, adding a BET inhibitor that further attenuates RAS activation and also upregulates IQGAP3 could result in the death of lymphoma cells that mainly thrive on the activation of these pathways. This was validated in our xenograft model, in which combined treatment with BET and PI3K inhibitors led to effective suppression of in vivo tumor growth. Moving forward, this leads to the conceptual rationale of employing concurrent PI3K and BET inhibition in the treatment of DLBCL, especially in the distinct PI3K-activated subtype. Our findings actually echo several previous reports focusing on adopting BET inhibitors and PI3K-targeting agents in the treatment of cancers (including DLBCL) [41,42,43], although the inter-collaborative activity between them differs across studies. However, the thorough mechanism on how IQGAP3 mitigates RAS activity and mediates DLBCL behaviors definitely warrants further investigation.

## 5. Conclusions

In conclusion, we unraveled a novel small GTPase-driven mechanism involved in DLBCL migration, a phenomenon unearthed through BET inhibition. We also identified IQGAP3, of which the expression is closely related to the activity of BET proteins, as a key indicator in further prognosticating a specific sub-group of DLBLCL patients who are otherwise deemed to have a very dismal outcome. These observations provide an important insight in our efforts to identify more effective ways of incorporating BET inhibitors into the therapeutic strategies of DLBCL.

## Figures and Tables

**Figure 1 cancers-13-05024-f001:**
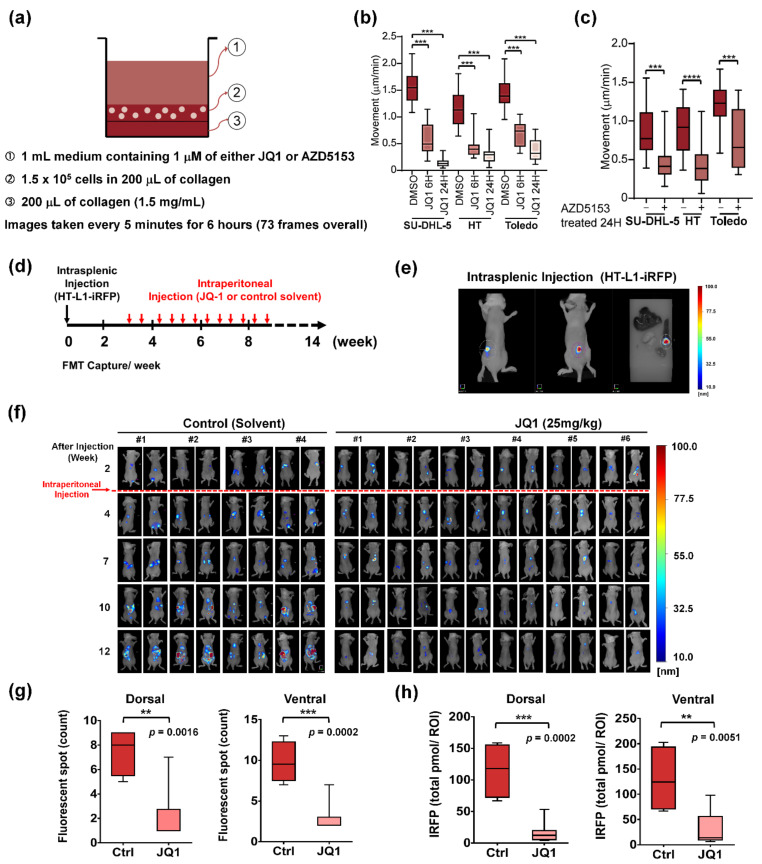
BET inhibition suppresses the migration and dissemination of DLBCL cells. (**a**) Illustration for migration assay. DLBCL cells, suspended in collagen solution, were deployed on top of another layer of collagen solution. After collagen solidification, 1 ml of medium containing JQ1 or AZD5153 was added and the cells were incubated for 6 or 24 hours. (**b,c**) Quantification of movement speeds of three DLBCL cell lines treated with either JQ1 or AZD5153 (*n* = 10). ** *p* < 0.01, *** *p* < 0.001, **** *p* < 0.0001, by Student’s t-test. (**d**) Schema for mice experiment. HT-L1 cells carrying an iRFP713 fluorescent reporter (HT-L1-iRFP) were injected into the spleens of 6-week-old SCID/beige mice. Intraperitoneal injection of JQ1 at the dose of 25 mg/kg was initiated at the end of week three. (**e**) Intrasplenic injection. Following injection, DLBCL tumors were assessed through the detection of infrared signals. Photos from a representative mouse are shown here (Control, *n* = 8; JQ1, *n* = 8). (**f**) Detection of near infrared signals. The near infrared signals in mice receiving treatment with either DMSO or JQ1 were taken at the indicated time points. (**g,h**) Enumeration of fluorescent tumor spots G and quantification of overall near infrared signals H in mice. Data collected through dorsal and ventral aspects of mice in Figure 1f are shown in box plots. Independent t-test was employed for statistical comparison.

**Figure 2 cancers-13-05024-f002:**
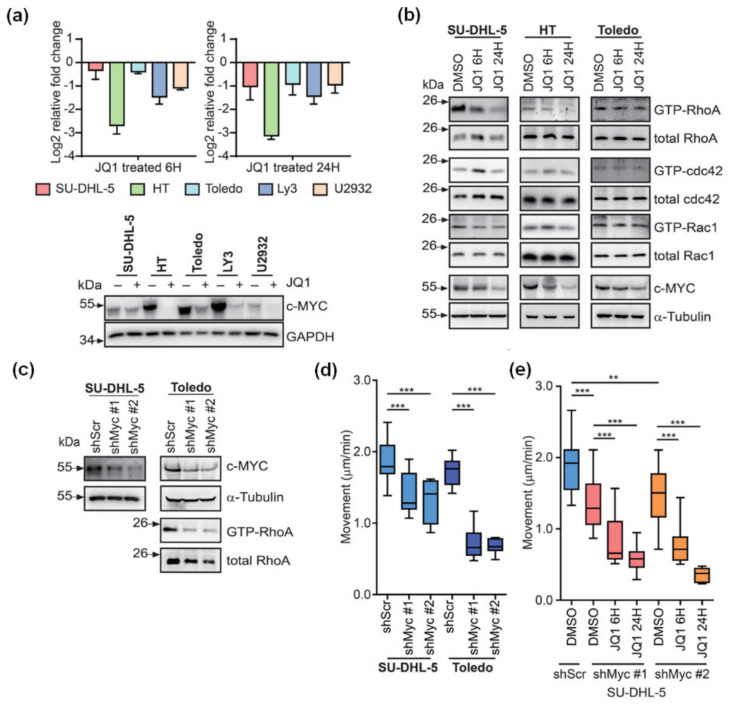
MYC suppression-mediated attenuated GTP-RhoA activity renders JQ1-treated DLBCL cells less mobile. (**a**) Measurement of the mRNA and protein expression levels of MYC in DLBCL cells treated with JQ1. Cells were incubated with 1 μM of JQ1 for indicated time periods. The level of *MYC* mRNA was quantified by qPCR (upper panel, *n* = 3), whereas the c-MYC protein level was assessed by Western blotting after 24-hour incubation with JQ1 (lower panel; the Western blots were repeated thrice, and the representative data are shown). Folds of mRNA expression were obtained through calculating the ratios between treated samples and untreated controls. (**b**) Western blots on small GTPase activity assays. After treatment with either DMSO or JQ1 for indicated time periods, GTP-bound RhoA, Cdc42, and Rac1 in whole cell lysates from SU-DHL-5, HT, and Toledo cells were analyzed after GST pull-down capture. The experiments were repeated thrice, and the representative data are shown. (**c**) Western blots on c-MYC and GTP-bound RhoA in HT and Toledo cells receiving either shRNA specific to Myc (shMyc; clones #1 and #2) or a scramble sequence (shScr). (**d**) Quantification of movement speeds of shMyc- or scramble control-treated DLBCL cells (*n* = 10). *** *p* < 0.001 by Student’s *t*-test. (**e**) Quantification of movement speeds of shMyc- or scramble control-treated HT cells in the presence of either DMSO or JQ1 for indicated time periods (*n* = 10). ** *p* < 0.01, *** *p* < 0.001, by Student’s *t*-test.

**Figure 3 cancers-13-05024-f003:**
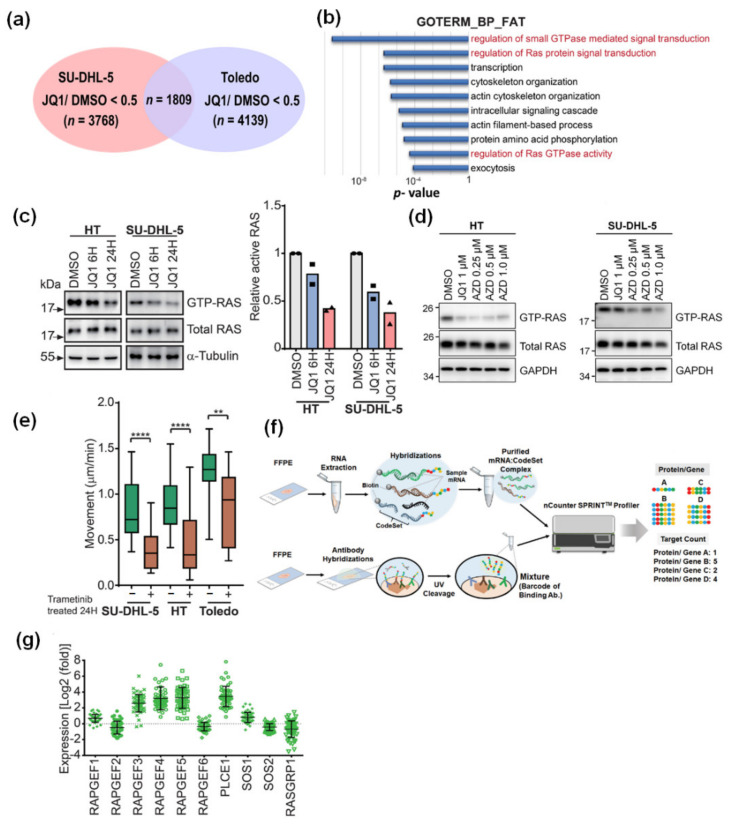
BET inhibition diminishes RAS activity in DLBCL cells. (**a**) A schema illustrating the mining process for the identification of relevant signal pathways affected by BET inhibition. SU-DHL-5 and Toledo cells were treated with either DMSO or JQ1 1μM for 8 hours and then subjected to RNA sequencing. The cutoff ratio of relative gene expression levels comparing JQ1- versus DMSO-treated cells was set at less than 0.5. (**b**) Gene Ontology-Gene function classification (Biological Process) database analysis. Several key signaling signatures differentially inactivated in JQ1-treated cells were listed. (**c,d**) Pull-down assay and Western blotting demonstrating RAS activity in DLBCL cells treated with BET inhibitors. Cells were treated with either JQ1 (concentration 1 μM) or AZD5153 at indicated concentrations for indicated time periods, and cell lysates were harvested for further analysis. Representative blots from three independent experiments were shown. (**e**) Quantification of movement speeds of three DLBCL cell lines treated with trametinib 100 nM (*n* = 10). ** *p* < 0.01, **** *p* < 0.0001, by Student’s *t*-test. Two biologically independent experiments were performed with consistent results, and the data shown here were from one representative experiment. (**f**) A schema demonstrating barcoded mRNA and protein analysis in tumor tissues from a cohort of DLBCL patients treated with immunochemotherapy. (**g**) Expression levels of several *RASGEF* genes in DLBCL samples. Folds of expression were obtained through calculating the ratios between DLBCL samples and normal controls. RASGEF: RAS guanine nucleotide exchange factors.

**Figure 4 cancers-13-05024-f004:**
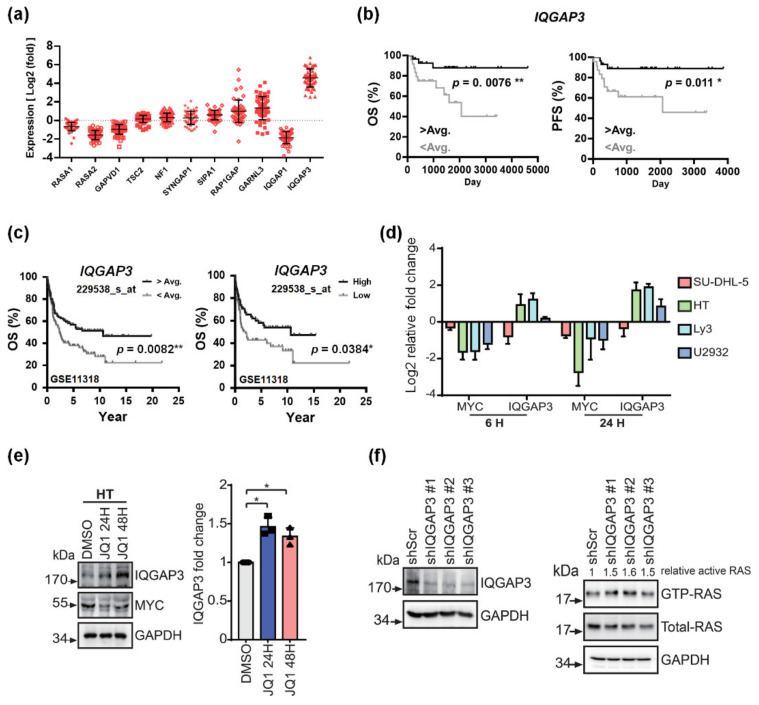
IQGAP3 mediates JQ1-induced RAS inactivation and portends a better survival in DLBCL patients. (**a**) Expression levels of several *RASGAP* genes in DLBCL samples. Folds of expression were obtained by calculating the ratios between DLBCL samples and normal controls. RASGAP: RAS GTPase-activating proteins. (**b**) Survival impacts of *IQGAP3* gene. DLBCL patients were stratified into two groups based on high (>average) or low (<average) expression of the *IQGAP3* gene. A Log-Rank test was employed to assess its prognostic significance on PFS and OS. IQGAP3: IQ Motif Containing Ras GTPase Activating Protein 3. OS: overall survival. PFS: progression-free survival. * *p* < 0.05, ** *p* < 0.01, by Student’s *t*-test. (**c**) External validation of prognostic significance of *IQGAP3* on survival outcome in DLBCL patients. GSE11318, a published data set from DLBCL mRNA series [17], was obtained for statistical analysis. Left panel: OS outcome in this cohort of 200 patients, with the subgroup stratification being based on the expression level of *IQGAP3* higher or lower than the average value. Right panel: OS outcome in 122 patients with either high or low levels of *IQGAP3* expression. Seventy-eight patients whose expression levels were within the range of average ± 0.5 x standard deviation were excluded from this cohort to better define the prognostic value of *IQGAP3*. * *p* < 0.05, ** *p* < 0.01, by Student’s *t*-test. (**d**) Measurement of the mRNA levels of *MYC* and *IQGAP3* in DLBCL cells treated with JQ1. Cells were incubated with 1 μM of JQ1 for indicated time periods. *n* = 3. Folds of the expression were obtained by calculating the ratios between treated samples and untreated controls. (**e**) Left: Western blots for examining the protein expression levels of MYC and IQGAP3 in DLBCL cells treated with JQ1. Cells were incubated with 1 μM of JQ1 for the indicated time periods. Right: quantification of the results from three independent experiments. (**f**) Western blots on RAS GTPase activity assays in HT cells receiving either shRNA specific to *IQGAP3* (shIQGAP3; clones #1, #2, and #3) or a scramble sequence (shScr). After treatment with shRNAs, GTP-bound RAS in whole cell lysates were analyzed after GST pull-down capture.

**Figure 5 cancers-13-05024-f005:**
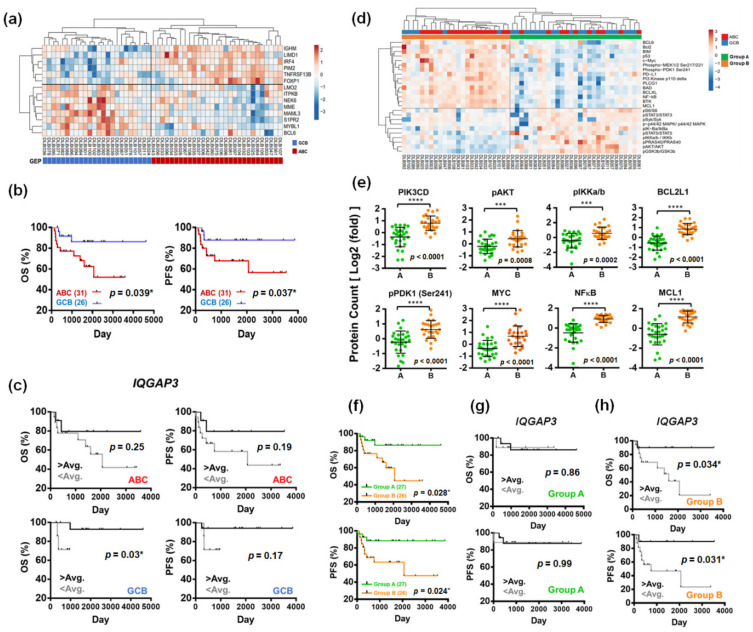
Proteomics-based re-categorization prognosticates DLBCL patients and reinforces the clinical significance of IQGAP3. (**a**) Employing gene expression profiling from the nCounter assay to stratify DLBCL based on their cells of origin (COO). ABC: activated B-cell-like; GCB: germinal center B-cell-like. (**b**) Survival outcome analysis based on COO stratification. Log-Rank test was employed to assess its prognostic significance on PFS and OS. * *p* < 0.05, by Student’s t-test. (**c**) The prognostic value of *IQGAP3* in individual subgroups of COO-stratified DLBCL. Patients were categorized based on high (>average) or low (<average) expressions of the *IQGAP3* gene. (**d**) Novel proteomics-based sub-classification of DLBCL. Some paired phosphorylated and respective total proteins were analyzed among the 35 proteins evaluated in the nCounter assay, and their expression levels were expressed as a phosphorylated form/total form. Unsupervised proteomic clustering analysis with Pearson’s correlation was performed. The first bar on the upper hand of the figure designates the COO of the tumors, whereas the second bar represents the two distinctly segregated groups (group A: green color; group B: orange color). (**e**) Comparison between group A and B on the expression levels of eight PI3K pathway-associated proteins. The data were obtained from nCounter analysis. All readings for each protein were subtracted by the maximum of the negative controls and normalized by the geometric mean of two internal controls (α-tubulin and histone H3). *** *p* < 0.001, **** *p* < 0.0001, by Student’s *t*-test. (**f**) Kaplan–Meier estimate of survival function in proteomics-stratified DLBCL patients. (**g,h**) Prognostic value of *IQGAP3* in individual DLBCL subgroups with discrepant PI3K signaling activity, stratified by proteomics. Patients were categorized based on high (>average) or low (<average) expressions of the *IQGAP3* gene.

**Figure 6 cancers-13-05024-f006:**
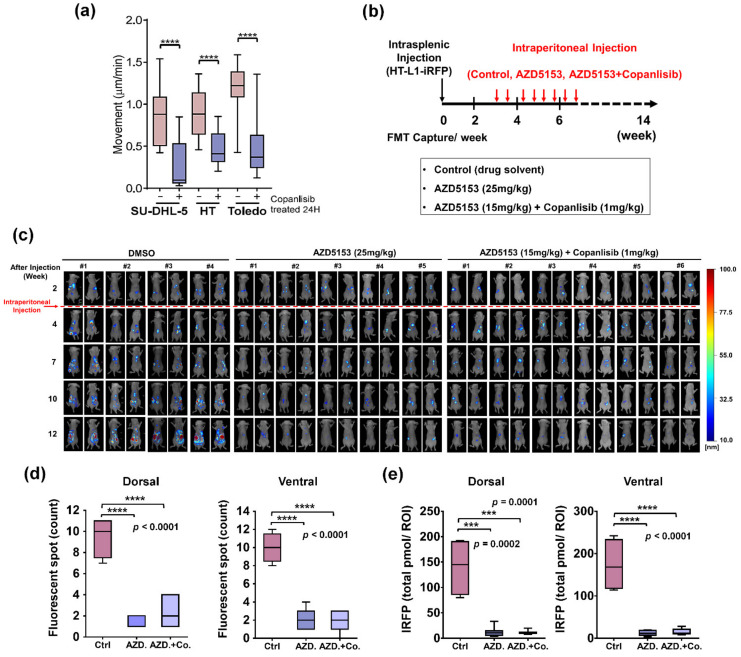
PI3K inhibitor attenuates the migration and dissemination of DLBCL. (**a**) Quantification of movement speeds of three DLBCL cell lines treated with copanlisib (0.5 μM) (*n* = 10). **** *p* < 0.0001, by Student’s *t*-test. (**b**) Schema for mice experiments. HT-L1-iRFP cells were injected into the spleens of 6-week-old SCID/beige mice. Intraperitoneal injection of drug solvent (*n* = 8), AZD5153 (25 mg/kg) (*n* = 8), or AZD5153 (15mg/kg) combined with copanlisib (1 mg/kg) (*n* = 8) twice weekly was initiated at the end of week 3. (**c**) Detection of near infrared signals. The fluorescent signals in mice receiving different treatment were taken at the indicated time points. The time indicated here represents weeks after intrasplenic injection. (**d,e**) Enumeration of fluorescent tumor spots (**d**) and quantification of overall near infrared signals (**e**) in mice. Data collected through dorsal and ventral aspects of mice in Figure 6c are shown in box plots. Independent *t*-test was employed for statistical comparison. *** *p* < 0.001, **** *p* < 0.0001, by Student’s *t*-test.

**Figure 7 cancers-13-05024-f007:**
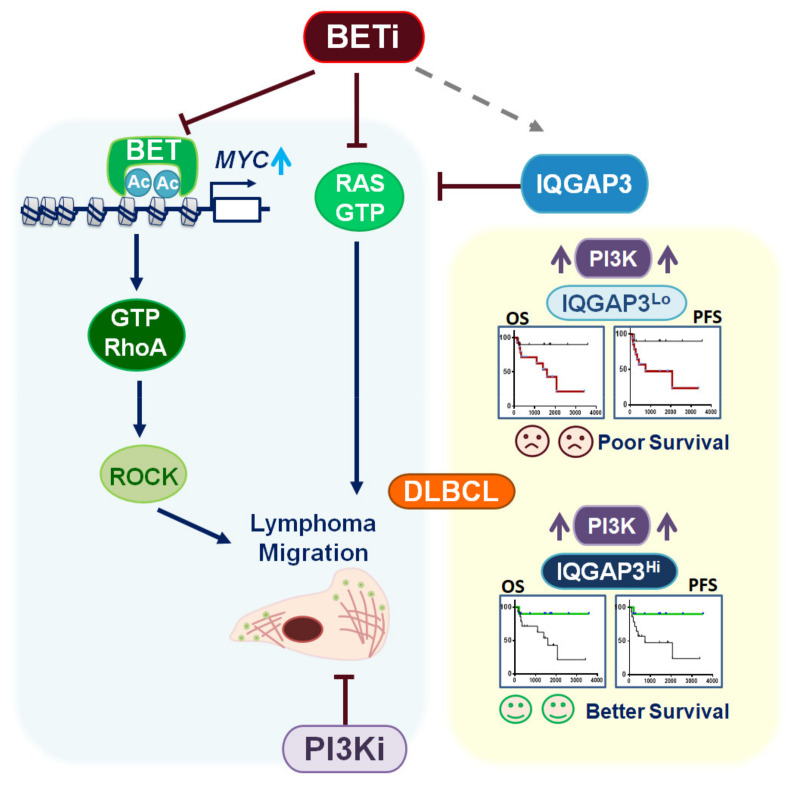
A schema demonstrating the molecular mechanism of BET and PI3K inhibition in DLBCL as well as the functional role and prognostic relevance of IQGAP3 in these tumors.

## Data Availability

All relevant data are available from the corresponding author upon reasonable request. The results of RNAseq in JQ1-treated cells are available at GEO under accession number GSE144821.

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
