# Peer review of "RAS Mediates BET Inhibitor-Endued Repression of Lymphoma Migration and Prognosticates a Novel Proteomics-Based Subgroup of DLBCL through Its Negative Regulator IQGAP3"

_cancers, 2021, doi:10.3390/cancers13195024_

Round 1
Reviewer 1 Report
This study by Chen et al. reports BET inhibitors' analysis in DLBCL cellular models, showing their putative value as a therapeutic approach in DLBLC.
The study shows two different parts, the first dealing with the role of BET inhibitors in DLBLC cells' mobility (amoeboid movements) through inhibition of RAS signaling and suppressing MYC-mediated RhoA activity. and also by the up-regulation of IQGAP3.
Then, they identify two DLBCL subtypes based on proteomics analysis. The one characterized by high PI3K activity showed a worse clinical course and more dismissal even for patients showing concurrent low IQGAP3 expression.
The study is of interest, well-written, and organized, including appropriate and exhaustive experiments and external validations.
There are, however, a few points that need clarifications.
The authors examined the impact of BET inhibition by RNA-seq in two cell lines, but how many replicates were analyzed? They found some relevant deregulated pathways in line with their other findings. Still, the statistical significance of these findings is not clear, and a fold change of only 0.5 (without replicates and only one cell line), with no other statistical test, does not seem strong enough.
In lines 351-354, the authors cited two papers from Schmitz et al. and Cahpuy et al. related to the new DLBCL genetic classifications. These references should also include the most recent ones by Lacy (Blood 2020) and Wright (Cancer Cell, 2020). Additionally, these are not, in fact, the best references for the COO classification. Those by Alizadeh (Nature 2020) and Scott (Blood 2014), for example, are more appropriate.
The real impact of IQGAP3 in the so-called group B is not clear. Are both variables independent? A multivariate analysis (or another adequate statistical test), could be informative.
Author Response
Cancers-1388839 – Response to reviewers
RAS mediates BET inhibitor-endued repression of lymphoma migration and prognosticates a novel proteomics-based subgroup of DLBCL through its negative regulator IQGAP3
Reviewer 1:
Reviewer 1 Comments for the Authors
This study by Chen et al. reports BET inhibitors' analysis in DLBCL cellular models, showing their putative value as a therapeutic approach in DLBLC. The study shows two different parts, the first dealing with the role of BET inhibitors in DLBLC cells' mobility (amoeboid movements) through inhibition of RAS signaling and suppressing MYC-mediated RhoA activity and also by the up-regulation of IQGAP3. Then, they identify two DLBCL subtypes based on proteomics analysis. The one characterized by high PI3K activity showed a worse clinical course and more dismissal even for patients showing concurrent low IQGAP3 expression. The study is of interest, well-written, and organized, including appropriate and exhaustive experiments and external validations.
There are, however, a few points that need clarifications.
Major comments
- The authors examined the impact of BET inhibition by RNA-seq in two cell lines, but how many replicates were analyzed? They found some relevant deregulated pathways in line with their other findings. Still, the statistical significance of these findings is not clear, and a fold change of only 0.5 (without replicates and only one cell line), with no other statistical test, does not seem strong enough.
Response: Thanks for the critical questions. As we mentioned in our article (Result section 3.3), the purpose of RNA sequencing was to try to figure out potential molecular mechanism of BET inhibitor-associated suppression of migration. Because of the restriction of our resources, we only analyzed TWO duplicate pairs of cells treated with either DMSO or JQ1 in individual cell lines, a huge limitation we fully agreed with our respectful reviewers. However, to minimize bias, we employed TWO cell lines (SU-DHL-5 and Toledo cells) to obtain genes that were consistently down-regulated upon JQ1 treatment (highlighted in the overlapping region in Figure 3A of the main manuscript). Importantly, results from the second independent experiment were not drastically different (data not shown).
Most of us would agree that correct identification of differentially expressed genes (DEGs) between specific conditions is a key in the understanding phenotypic variation. However, there has been no consensus on the most appropriate protocol, cutoff value, or pipeline for identifying DEGs from RNA-Seq data [1-3]. We understand that data obtained from small sample size like ours should be interpreted with caution. Therefore, we did not elaborate much on the RNA-Seq results. Instead, we explored through small GTPase-mediated signal transduction signature, a key pathway identified in our transcriptomic analysis employing JQ1-treated cells (p < 0.0001). Our subsequent in vitro experiments and clinical sample analysis confirmed the pathognomonic as well as prognostic role of this pathway in DLBCL.
In brief, our RNA-seq data were imperfect, but such an analysis helped us unearth small GTPase-mediated signal transduction signatures in DLBCL.
- In lines 351-354, the authors cited two papers from Schmitz et al. and Cahpuy et al. related to the new DLBCL genetic classifications. These references should also include the most recent ones by Lacy (Blood 2020) and Wright (Cancer Cell, 2020). Additionally, these are not, in fact, the best references for the COO classification. Those by Alizadeh (Nature 2020) and Scott (Blood 2014), for example, are more appropriate.
Response: Thanks so much for the invaluable comments and suggestion. We apologize for not more properly citing relevant information. These important references were added to our revised manuscript accordingly.
- The real impact of IQGAP3 in the so-called group B is not clear. Are both variables independent? A multivariate analysis (or another adequate statistical test), could be informative.
Response: Thanks for the indispensable reminder. In an effort to perform mulvariate analysis, we incorporated four parameters – cell of origin (COO; ABC vs. GCB), nCounter-based grouping (A vs. B), expression levels of IQGAP3 (high vs. low), and international prognostic index (IPI; low/low-intermediate vs. high/high-intermediate) – into outcome analysis regarding progression-free survival (PFS) and overall survival (OS) in our patient cohort. The results could be listed as follows (Table 1):
Table 1. Multivariate analyses of some key variables on survival outcome
|
|
PFS |
OS |
||
|
|
HR |
p value |
HR |
p value |
|
COO (GCB vs. ABC) |
0.342 |
0.224 |
0.568 |
0.662 |
|
nCounter grouping (B vs. A) |
3.168 |
0.154 |
7.596 |
0.077 |
|
IQGAP3 (high vs. low) |
0.153 |
0.027 |
0.000 |
0.941 |
|
IPI risk (H/HI vs L/LI) |
11.984 |
0.002 |
17.032 |
0.002 |
* IPI risk: H/HI: high and high-intermediate; L/LI: low and low-intermediate
From the table, it seemed that IPI risk category was the only factor affecting both PFS and OS, whereas a high IQGAP3 level foretold an excellent PFS outcome but did not play major role in predicting OS outcome. However, nCounter-based gene expression profiling for molecular classification of COO could not be utilized to prognosticate both the PFS and OS in our DLBCL patients. This noticeably contradicts what we know about the role of COO in DLBCL, as numerous studies have confirmed the poor prognosis of the ABC subtype among DLBCL patients. The problem here apparently lies in too many factors (four, all of which are considered important) being incorporated into a relatively small cohort (53 cases) of DLBCL patients for statistical comparison, a glaring limitation of our current study. Consequently, it’s difficult to appraise the prognostic significance of nCounter grouping and IQGAP3 expressional status in the multivariate analysis employing our patient cohort.
However, we still think our results have some merits. First of all, we did validate the predictive value of a high IQGAP3 expression level in an independent cohort of DLBCL patients (Figure 4C in the main manuscript, GSE11318) [4]. More importantly, we demonstrated that IQGAP3 could affect RAS-GTPase activity (Figure 4f in the main manuscript), a signaling pathway proved to be vital in lymphoma migration and dissemination. In other words, the role of IQGAP3 in DLBCL is meticulously supported by our in vitro experiments. While we are very happy to report this novel finding, we also think it definitely warrants further validation from other groups.
References:
- Costa-Silva, J.; Domingues, D.; Lopes, F.M. RNA-Seq differential expression analysis: An extended review and a software tool. PLoS One 2017, 12, e0190152, doi:10.1371/journal.pone.0190152.
- Zhang, Z.H.; Jhaveri, D.J.; Marshall, V.M.; Bauer, D.C.; Edson, J.; Narayanan, R.K.; Robinson, G.J.; Lundberg, A.E.; Bartlett, P.F.; Wray, N.R.; et al. A comparative study of techniques for differential expression analysis on RNA-Seq data. PLoS One 2014, 9, e103207, doi:10.1371/journal.pone.0103207.
- Rapaport, F.; Khanin, R.; Liang, Y.; Pirun, M.; Krek, A.; Zumbo, P.; Mason, C.E.; Socci, N.D.; Betel, D. Comprehensive evaluation of differential gene expression analysis methods for RNA-seq data. Genome Biol 2013, 14, R95, doi:10.1186/gb-2013-14-9-r95.
- Lenz, G.; Wright, G.W.; Emre, N.C.; Kohlhammer, H.; Dave, S.S.; Davis, R.E.; Carty, S.; Lam, L.T.; Shaffer, A.L.; Xiao, W.; et al. Molecular subtypes of diffuse large B-cell lymphoma arise by distinct genetic pathways. Proceedings of the National Academy of Sciences of the United States of America 2008, 105, 13520-13525, doi:10.1073/pnas.0804295105.

Reviewer 2 Report
Chih-Cheng Chen and co-authors Manuscript entitled “ RAS mediates BET inhibitor-endued repression of lymphoma 2 migration and prognosticates a novel proteomics-based 3 subgroup of DLBCL through its negative regulator IQGAP3” is very well written and experiments are performed meticulously. Authors have connected the study very well and have shown the inhibition of MYC-RHOa GTP after using BET/JQ1 inhibitors and transcriptomic data reveled IQGAP3 as prognostic marker in DLBSL. This paper has potential to be accepted after minor revisions. Please address the followings
- Mention the drug dose of JQ1 and AZD5153 in Fig-1
- There is no consistency of cell lines used, In figure 1, LY3 and and U2392 cell lines were not treated with JQ1 or AZD5153.
- Provide the mutation status of the all-cell lines used in this study.
- Authors need to explore the transcriptomic data further by using GSEA or DAVID and report the other pathways if found.
- Authors should also show the correlation of transcriptomic data and proteomic data.
- Check line 292, remove in ” peculiarly up-regulated in in JQ1-treated DLBCL cells”
Author Response
Cancers-1388839 – Response to reviewers
RAS mediates BET inhibitor-endued repression of lymphoma migration and prognosticates a novel proteomics-based subgroup of DLBCL through its negative regulator IQGAP3
Reviewer 2:
Reviewer 2 Comments for the Authors
Chih-Cheng Chen and co-authors Manuscript entitled “RAS mediates BET inhibitor-endued repression of lymphoma migration and prognosticates a novel proteomics-based subgroup of DLBCL through its negative regulator IQGAP3” is very well written and experiments are performed meticulously. Authors have connected the study very well and have shown the inhibition of MYC-RHOa GTP after using BET/JQ1 inhibitors and transcriptomic data reveled IQGAP3 as prognostic marker in DLBSL. This paper has potential to be accepted after minor revisions. Please address the followings
Major comments
- Mention the drug dose of JQ1 and AZD5153 in Fig-1
Response: Thanks for providing us a chance for further clarification. The doses of JQ1 and AZD5153 for the migration assay (both 1 mM) were illustrated in Figure 1a of the main manuscript, whereas the dose of JQ1 for the in vivo mice experiment (25 mg/kg) was depicted in Figure 1f of the main manuscript.
- There is no consistency of cell lines used. In figure 1, LY3 and U2392 cell lines were not treated with JQ1 or AZD5153.
Response: Thanks very much for asking this key question. In this paper, we tried to make our work as convincing and reliably reproducible as possible. That’s why we chose five DLBCL cell lines for our experiments. However, as my highly esteemed reviewers and editors would agree, these meticulously performed experiments were indeed exhaustive. We tried to compromise a bit and selected SU-DHL-5, HT, and Toledo cells as the backbone of most of our in vitro experiments. HT-L1, a derivative subline of HT cells which exhibited enhanced in vivo tumorigenicity in our previous work [1], was accordingly employed in the animal model. Both LY3 and U2392 cell lines were simply used to validate the changes in the expressional levels of MYC and IQGAP3 upon JQ1 treatment. With the migration assay more complicated and time-consuming, coupled with the consistent findings across three different DLBCL cell lines (Figure 1b and 1c), we did not incorporate LY3 and U2392 in these experiments. However, if it is deemed necessary, we are willing to repeat the migration assay with these cells.
- Provide the mutation status of the all-cell lines used in this study.
Response: Thanks for the kind reminder. Attached please find the relevant information of the cell lines used in the study.
Table 1. Genetic, cytogenetic, and immunophenotypic characteristics of DLBCL cells
|
Cell line (COO) |
Genetic status |
|
HT (GCB) |
l Immunology: CD19 +; CD20 +; CD21 +; CD22 +; Hle-1 +; HLA DQ +; HLA DR +; CD25 -; T cell receptor (TCR) -; express mRNA for both IgM and kappa l Cytogenetics: human near-diploid karyotype with 3% polyploidy - 46(42-46)<2n>XY, +2, der(2;4)(p10;q10), der(2)(del)(2)(p11?q21), dup(10)(q11q22-23), dup(11)(?q23qter) - no IGH-BCL2 rearrangement detected l Virus: EBV -, HBV -, HCV -, HIV-1 -, HIV-2 -, HTLV-I/II -, MLV -, SMRV – l Mutations: https://depmap.org/portal/cell_line/ACH-000914?tab=mutation |
|
OCI-LY3 (ABC) |
l Immunology: CD3 -, CD10 -, CD13 -, CD19 (+), CD20 +, CD34 -, CD37 +, cyCD79a +, CD80 +, CD138 -, HLA-DR +, sm/cyIgG +, sm/cyIgM -, sm/cykappa -, sm/cylambda +; in the referenced paper this cell line is described as cykappa + l Cytogenetics: human flat-moded hypertriploid karyotype; 72-77<3n>XXYY, +1, +9, -10, +13, +14, -17, +19, +20, +22, der(1)t(1;17)(p13;q12)x2, der(4)t(4;18)(q31;q21)x2, del(6)(q13)x2, der(6)t(6;6)(p24;q12), der(7)t(6;7)(p24;p22), der(14)t(14;19)(q32;q13.3)x2, del(18)(q21), der(19)t(4;19)(q21;q13)t(4;18)(q31;q21)x2, der(19)t(14;19), dup(20)(q11q13)x2; sdl with der(6)t(6;12)(p21;q21), der(7)t(5;7)(?p15;p24) etc; resembles published karyotypes; carries cryptic t(14;19) with rearrangement of IGH and SPIB, and t(4;18) with copy number amplification of the BCL2 region l Virus: HBV -, HCV -, HIV-1 -, HIV-2 -, MLV – l Mutations: https://depmap.org/portal/cell_line/ACH-000158?tab=mutation |
|
SU-DHL-5 (GCB) |
l Immunology: CD3 -, CD10 +, CD13 -, CD19 +, CD20 +, CD37 +, cyCD79a +, CD80 -, HLA-DR +, sm/cyIgG -, sm/cyIgM +, sm/cykappa -, sm/cylambda + l Cytogenetics: human hyperdiploid karyotype with 1.5% polyploidy - 47(41-48)<2n>XX, +12, del(6)(q13), del(12)(q13) - sideline with del(6)x2 - matches published karyotype l Virus: EBV -, HBV -, HCV -, HIV-1 -, HIV-2 -, HTLV-I/II -, MLV -, SMRV – l Mutations: https://depmap.org/portal/cell_line/ACH-000660?tab=mutation |
|
Toledo (GCB) |
l Immunology: CD10 +, CD19 +, CD20 +, CD38 +, CD23 -, CD39 – l Cytogenetics: The karyotype does exhibit multiple chromosomal aberrations; The cells do not express surface or cytoplasmic immunoglobulin; carries t(8;14) effecting MYC-IGH rearrangement; carries t(14;18) effecting IGH-BCL2 fusion l Virus: EBV – l Mutations: https://depmap.org/portal/cell_line/ACH-000285?tab=mutation |
|
U-2932 (ABC) |
l Immunology: CD3 -, CD10 +, CD13 -, CD19 +, CD20 +, CD37 +, CD38 +, cyCD79a +, CD80 +, CD138 +, HLA-DR +, sm/cyIgG -, sm/cyIgM +, sm/cykappa +, sm/cylambda - l Cytogenetics: human polyclonal hypodiploid karyotype with 20% polyploidy; 45(43-46)<2n>, XX, add(X)(q22), add(1)(q24), der(3)ins(3;18)(q27;q2?q2?)hsr(18)(q21), add(5)(q32), der(6)t(6;18)(p24;?q23)del(6)(q13), der(10)t(10;14)(q24;q23), der(11)t(1;11)(q25;q2?), der(14)t(3;14)(q27;p11), der(18)t(1;18)(q21;q21), der(18)t(3;18)(q2?;q27), add(19)(p13), del(19)(q13); sdl with del(2)(q11), t(4;15)(q22;q14), add(7)(q21), del(13)(?q21), der(13)add(13)(p11)add(13)(q32); extensive genomic amplification of BCL2 region; resembles published karyotype l Virus: EBV -, HBV -, HCV -, HIV-1 -, HIV-2 -, HTLV-I/II -, MLV -, SMRV – l Mutations: https://cancer.sanger.ac.uk/cosmic/sample/overview?id=1945196#variants |
- Authors need to explore the transcriptomic data further by using GSEA or DAVID and report the other pathways if found.
Response: Thanks so much for the precious comment and suggestion. In our submitted work, we performed Gene Ontology pathway analysis and identified several key signaling signatures that were differentially inactivated in JQ1-treated cells. We selected several major key pathways and highlighted them in Figure 3b of the main manuscript. The remaining pathways with statistical significance (defined as a p-value < 0.1) were listed in the supplementary file (Supplemental Table 7). These pathways included regulation of I-kB kinase/NF-kB cascade, regulation of programmed cell death, regulation of MAPKKK cascade, and so on. To confirm the reproducibility of these data, we also employed DAVID to analyze those down-regulated genes. The table (Table 2) and figure (Figure 1) below demonstrated results of the major pathways of interests obtained from both GO and DAVID analyses, which we listed side by side. Based on these findings, it is reasonable to believe the identified major pathways are agreeable when either analysis is used. Other pathways identified in DAVID analysis are attached in a separate excel file, which listed only those with a FDR value and a p-value both less than 0.05.
Table 2. Comparison on key pathways identified through GO and DAVID analyses.
|
Gene Ontology Analysis / Term |
P-Value |
DAVID Analysis / Term |
P-Value |
|
regulation of small GTPase mediated signal transduction |
2.5E-08 |
positive regulation of GTPase activity (GO:0043547) |
4.77E-09 |
|
regulation of Ras protein signal transduction |
1.9E-05 |
Ras protein signal transduction (GO:0007265) |
3.22E-04 |
|
transcription |
1.9E-05 |
Transcription |
1.18E-08 |
|
cytoskeleton organization |
4.3E-05 |
cytoskeleton organization (GO:0007010) |
9.87E-05 |
|
actin cytoskeleton organization |
4.7E-05 |
actin cytoskeleton organization (GO:0030036) |
4.67E-05 |
|
intracellular signaling cascade |
1.3E-04 |
intracellular signal transduction (GO:0035556) |
1.56E-10 |
|
actin filament-based process |
2.0E-04 |
actin filament-based process (GO:0030029) |
1.43E-04 |
|
protein amino acid phosphorylation |
2.5E-04 |
protein phosphorylation (GO:0006468) |
3.62E-06 |
|
regulation of Ras GTPase activity |
4.9E-04 |
Ras GTPase binding (GO:0017016) |
0.00302 |
|
exocytosis |
7.9E-04 |
Exocytosis |
0.0036 |
Please see figure in attached file
Figure 1. Comparison on key pathways identified through GO and DAVID analyses.
- Authors should also show the correlation of transcriptomic data and proteomic data.
Response: Thanks so much for the indispensable suggestion. It’s noteworthy that many of the studied proteins in the nCounter assay were functional proteins in phosphorylation form. As a result, only few genes had concurrent assessment in mRNA and protein (non-phosphorylated) levels, which made adequate correlation between transcriptomic and proteomic data less feasible. Nevertheless, we still did some analyses in certain genes and observed various degrees of positive correlation between the mRNA and protein levels. Two examples (BCL2 and BAD) are demonstrated below (Figure 2):
(A) (B)
Please see figure in attached file
Figure 2. Correlation between mRNA and protein levels in two representative genes. (A) BCL2: strong positive correlation (B) BAD: modest positive correlation
- Check line 292, remove in ” peculiarly up-regulated in in JQ1-treated DLBCL cells”
Response: Thanks for bringing up this issue for discussion. In Figure 4a, we employed the nCounter transcriptomic analysis to identify IQGAP3 as the most prominently over-expressed RASGAP genes in clinical DLBCL samples. We looked back into DLBCL cells and found that IQGAP3 was one of few key RASGAP genes that were up-regulated upon JQ1 treatment. These findings brought up the possibility that clinical findings could well be validated through in vitro experiments, which in turn led to subsequent analysis. Therefore, we think that if we can retain the sentence here, our intention and findings could be more comprehensible to the readers. However, we do appreciate your great suggestion.
References:
- Pan, Y.R.; Chen, C.C.; Chan, Y.T.; Wang, H.J.; Chien, F.T.; Chen, Y.L.; Liu, J.L.; Yang, M.H. STAT3-coordinated migration facilitates the dissemination of diffuse large B-cell lymphomas. Nature communications 2018, 9, 3696, doi:10.1038/s41467-018-06134-z.
